

# The mechanism of static postural control in the impact of lower limb muscle strength asymmetry on gait performance in the elderly

Beili Si[1], Hao Zhu[2], Xinmei Wei[3], Shun Li[1] and Xueping Wu[1]

[1] School of Physical Education, Shanghai University of Sport, Shanghai, China
[2] School of Psychology, Shanghai University of Sport, Shanghai, China
[3] School of Economics and Management, Shanghai University of Sport, Shanghai, China

## ABSTRACT

**Background:** Abnormal gait is prevalent among the elderly population, leading to reduced physical activity, increased risk of falls, and the potential development of dementia and disabilities, thus degrading the quality of life in later years. Numerous studies have highlighted the crucial roles of lower limb muscle strength asymmetry and static postural control in gait, and the reciprocal influence of lower limb muscle strength asymmetry on static postural control. However, research exploring the interrelationship between lower limb muscle strength asymmetry, static postural control, and gait performance has been limited.

**Methods:** A total of 55 elderly participants aged 60 to 75 years were recruited. Isokinetic muscle strength testing was used to assess bilateral knee extension strength, and asymmetry values were calculated. Participants with asymmetry greater than 15% were categorized as the Asymmetry Group (AG), while those with asymmetry less than 15% were classified in the Symmetry Group (SG). Gait parameters were measured using a plantar pressure gait analysis system to evaluate gait performance, and static postural control was assessed through comfortable and narrow stance tests.

**Results:** First, participants in the AG demonstrated inferior gait performance, characterized by slower gait speed, longer stance time and percentage of stance time in gait, and smaller swing time and percentage of swing time in gait. Spatial-temporal gait parameters of the weaker limb tended to be abnormal. Second, static postural control indices were higher in AG compared to SG in all aspects except for the area of ellipse during the comfortable stance with eyes open test. Third, abnormal gait parameters were associated with static postural control.

**Conclusion:** Firstly, elderly individuals with lower limb muscle strength asymmetry are prone to abnormal gait, with the weaker limb exhibiting poorer gait performance. Secondly, lower limb muscle strength asymmetry contributes to diminished static postural control in the elderly. Thirdly, the mechanism underlying abnormal gait in the elderly due to lower limb muscle strength asymmetry may be linked to a decline in static postural control.

Corresponding author
Xueping Wu, wuxueping@sus.edu.cn

## INTRODUCTION

The prevalence of gait disorders increases with age, ranging from about 10% in people aged 60 to 69 years to over 60% in those over 80 years of age (*Mahlknecht et al., 2013*). Abnormal gait usually manifests as slower gait velocity and cadence, shorter step length, smaller first double contact point, larger last double contact point, and a larger proportion of stance and swing phase time (*Hollman, McDade & Petersen, 2011*; *Moreira, Sampaio & Kirkwood, 2015*; *Raccagni et al., 2020*; *Niederer et al., 2021*; *Indelicato et al., 2022*).
The emergence of abnormal gait causes the elderly to reduce physical activity and increases the risk of falling (*Paterson, Hill & Lythgo, 2011*; *Marini et al., 2022*). Physical activity reduction may accelerate dementia (*Livingston et al., 2020*), thereby causing the elderly to lose their independence in physiological functions and quality of life during their later years (*Guralnik et al., 2000*; *Livingston et al., 2020*; *Qiao et al., 2022*). Therefore, the diagnosis of gait abnormalities and their aetiologies at the earliest possible timepoint is essential to improve the quality of life of the elderly.

Lower limb muscle strength asymmetry is an important factor in the pathogenesis of gait disorders in the elderly (*Laroche, Cook & Mackala, 2012*). This asymmetry refers to unequal muscle strength of contralateral limbs (*Parkinson et al., 2021*) and is related to both the dominant limb (*Lanshammar & Ribom, 2011*) and its pathology (*Suetta et al., 2007*). The general consensus is that a bilateral lower limb strength difference of 10–15% or greater is considered problematic (*Rohman, Steubs & Tompkins, 2015*; *Bishop, Turner & Read, 2018*; *Parkinson et al., 2021*). Among the components of lower limb strength, knee extension strength is particularly critical for maintaining stable and efficient gait in the elderly. During the initial stance phase, the quadriceps' activation is crucial for absorbing impact forces and stabilizing the body. As the stance phase progresses, the quadriceps maintain their function, ensuring knee stability until the foot lifts off the ground and transitions into the swing phase (*Ellis, Sumner & Kram, 2014*; *Ogaya et al., 2017*). The disparity in muscle strength between the lower limbs, especially in knee extension, often intensifies with age (*Goodpaster et al. 2006*; *Perry et al., 2007*; *Carabello et al., 2010*), leading to impaired balance and coordination, and consequently, gait abnormalities (*Trzaskoma, Tihanyi & Trzaskoma, 2010*; *Laroche, Cook & Mackala, 2012*; *Bond et al., 2017*). Despite the recognized impact, the mechanism by which lower limb muscle strength asymmetry causes gait abnormalities in the elderly has not been clarified. Therefore, the exploration of the impact of lower limb muscle strength asymmetry on gait among the elderly is of crucial importance.

Additionally, gait performance is a complex coordination of musculoskeletal, neurological, sensory, and cognitive systems (*Aboutorabi et al., 2016*), with postural control being intrinsically linked to this dynamic interaction (*Brincks et al., 2017*). Deterioration in postural control cascades into compromised gait dynamics, exemplified by reduced walking speed and altered temporal parameters (*Xie et al., 2017*; *Chung et al., 2022*). Specially, poor

static postural control will lead to slower gait velocity and increased support and swing times (Chung et al., 2022). Debates in the sports medicine and rehabilitation on the relationship between muscle strength asymmetry, postural control, and gait performance in the elderly is currently limited. Although Chung et al.'s (2022) study found that muscle strength and postural control can affect gait performance, Forte et al. (2014) observed that subjects with less body sway can complete complex gait tasks faster, and that postural control impacts gait performance through the lower limbs. Muscle strength plays an important role in maintaining the balance relationship during walking (Koda et al., 2018; Drijkoningen et al., 2015) found that the degree of muscle strength asymmetry was directly related to impaired static postural control and gait variability; however, the study focused on young people with brain injuries or subjects regardless of age. To our knowledge, research has not been conducted on the relationship between lower limb muscle strength asymmetry, static postural control, and gait performance in the elderly population. Therefore, to fill this research gap, this study explored how lower limb muscle strength asymmetry affects static postural control and gait performance in the elderly. Our findings may inform the further development of exercise plans for the elderly to improve gait disorders.

The purpose of this study was to explore the influence of knee extension force asymmetry on static postural control and spatiotemporal gait parameters in the elderly, and to further analyse whether static postural control induced by lower limb muscle strength asymmetry is linked to gait performance. We hypothesized that the asymmetry in knee extension strength would correlate with diminished static postural control and increased gait variability, and that static postural control induced by lower limb muscle strength asymmetry was linked to gait performance in older adults.

## MATERIALS AND METHODS

### Participants

Using G*Power (version 3.1) for *a priori* analysis, it was determined that a minimum total sample size of 34 individuals is necessary to achieve a statistical power of 0.80 for a medium effect size ($f = 0.25$) in a $2 \times 2$ repeated measures ANOVA (Cohen, 1992), with a significance level of $\alpha = 0.05$. Participant recruitment was conducted within communities in Yangpu District, Shanghai, and a total of 55 participants (23 males and 32 females) were enrolled, which met the study's sample size requirements. The inclusion criteria included: 1) aged 60 to 75 years; 2) capable of independent living and unassisted walking; 3) no history of lower limb surgery; 4) no lower limb pain in the week preceding the test. Participants were excluded if they had: 1) severe cardiovascular or cerebrovascular diseases; 2) neurological disorders such as Parkinson's disease, spinal conditions, or stroke; 3) visual or auditory impairments; 4) a history of severe lower limb injury within the past 6 months; 5) pain during the maximum strength test that prevented test completion. Informed consent was obtained from all participants after a thorough explanation of the potential risks and discomforts. The participants were divided into two groups: symmetrical strength (SG, $n = 29$) and asymmetrical strength (AG, $n = 26$), based on a 15%

asymmetry threshold. The study protocol underwent scrutiny and received endorsement from the Ethics Committee of Shanghai University of Sport (No. 102772022RT123).

## Lower limb muscle strength asymmetry

The "gold standard" for assessing human muscle strength (*Kambič, Lainščak & Hadžić, 2020*), Isokinetic muscular strength testing equipment (CON-TREX MJ, Physiomed elektromedizin AG, Schnaittach, Germany) was used to evaluate bilateral knee extensor strength. Before commencing the test, participants were instructed to engage in a 5-min warm-up session on a power cycle, followed by dynamic stretching of their lower limb muscles. Subsequently, the participants assumed a seated posture, and the torso and non-examined limb were fastened to the testing chair to ensure safety. Prior to commencing the formal test, participants were required to acquaint themselves with the range of motion and successfully complete five tests at maximal effort. As follows, the angular velocity of movement was set at 60°/s, and the range of motion was measured at 90°. The isokinetic concentric mode was used for testing. The measurable parameter was peak torque (Nm). The lower limb muscle strength asymmetry calculation was based on *Laroche, Cook & Mackala*'s *(2012)* work and set the threshold at 15% based on most scholarly definitions of asymmetry (*Barber et al., 1990*; *Perry et al., 2007*; *Parkinson et al., 2021*). More precisely, lower limb muscle strength asymmetry was defined as a difference of peak torque of knee extension of more than 15%.

$$\text{Asymmetry } (\%) = \frac{|(\text{Strength of Left Limb } - \text{ Strength of Right Limb })|}{\text{Strength of Strong Limb}} * 100\%$$

## Static postural control

The ability to maintain static postural control was assessed by using a force platform (Physio sensing, Sensing Future Technologies, Coimbra, Portugal). Participants were instructed to maintain a straight-ahead binocular gaze and to place both palms on their hips throughout the test. Each assessment protocol incorporated eyes open (EO) and eyes closed (EC) visual conditions, and comfortable standing (CS) and narrow standing (NS) widths (*Riemann & Piersol, 2017*; *Scoppa et al., 2017*). The sway velocity index (SVI) was determined once all protocol conditions had been met. The natural logarithm function was applied to normalise the result of dividing the participant's height by the mediolateral velocity to calculate SVI (*Riemann et al., 2018*). Mediolateral velocity was calculated by dividing the displacement of the centre of pressure during the trial by the duration of the trial in milliseconds. The sway displacement in the mediolateral direction was calculated for every trial during every acquisition (100 acquisitions per second) and the software calculated the mean result at the conclusion. Moreover, the area of the ellipse (AOE) was calculated by utilising the trajectory of the pressure centre for each test.

## Gait performance

Gait analysis utilised the plantar pressure gait assessment system (MedTrack, Xinkang Biomedical Technology Co., Hangzhou, CHN). The system comprises a computer, a

pressure-sensitive walkway, and specialised analysis software, enabling the uninterrupted collecting of gait metrics at a sample frequency of 400 Hz. The supplementary computer accurately identifies and assesses the participant's locomotion data to generate spatiotemporal parameters with the assistance of its integrated software. The dimensions of the walkway are 7.2 m in length and 1.0 m in width, with buffer zones of 1 m at both ends. The equipment used in this study is distinct from that in previous research, as it independently calculates the gait parameters for both the left and right limbs of subjects (with data analysis distinguishing between the strong and weak limbs), thereby facilitating a more detailed examination of the influence of muscle strength asymmetry on the spatiotemporal gait parameters in older adults.

Participants were instructed to remove their shoes and walk barefooted at a natural speed for the entire test duration, starting 1 m from the walkway. The test concluded when the participant reached the endpoint. All 14 gait parameters—step length; stride length; step time; stride time; stance time; swing time; single support time (SST); double support time (DST); percentage of stance time in gait cycle (STT%); percentage of swing time in gait cycle (SWT%); percentage of SST in gait cycle (SST%); percentage of DST in gait cycle (DST%)—were used for analysis. All the above parameters except for gait velocity and cadence included both the strong and weak limbs.

### Statistical analyses

Experimental data were analysed with SPSS software (IBM, Armonk, NY, USA). A repeated measures analysis of variance (ANOVA) with a $2 \times 2$ design was used to investigate intergroup variations in data pertaining to comfortable/narrow stance conditions in the static postural control test. In addition, a comparative analysis was performed to examine the disparities in data pertaining to the strong and weak limbs during the gait performance test. Ultimately, Pearson correlation analysis was utilised to evaluate the association between the notable disparities revealed in the static postural control test and gait parameters. A significance level of $P < 0.05$ was established.

## RESULTS

### Participant characteristics

The demographic characteristics of the two participant categories are detailed in Table 1. Age, gender, height, weight, body mass index (BMI), preferred limb side, and the strong and weak knee extensor strength were statistically similar between the two cohorts, suggesting that all participants were comparable.

### Gait performance

The independent samples t-test results comparing gait velocity and cadence are presented in Table 2. Gait velocity ($t_{(53)} = 3.557$, $P < 0.01$, Cohen's d = 0.97) and cadence ($t_{(53)} = 2.371$, $P < 0.05$, Cohen's d = 0.65) were significantly slower in AG than in SG.

Using SG and AG as between-group factors, and left and right limb lateralization as within-group factors, a repeated measures analysis of variance was employed to examine differences in spatiotemporal gait parameters between the strong and weak limbs within

**Table 1 The characteristics of participants in each group.**

|  | SG | AG | $t/\chi^2$ | $P$ |
|---|---|---|---|---|
| Age (y) | 67.97 ± 3.78 | 66.00 ± 3.90 | 1.896 | 0.063 |
| Gender (male/female) | 13/16 | 10/16 | 0.228 | 0.633 |
| Height (cm) | 164.83 ± 8.02 | 165.08 ± 8.40 | −0.113 | 0.911 |
| Mass (kg) | 64.50 ± 10.66 | 64.96 ± 9.86 | −0.165 | 0.870 |
| BMI (kg/m²) | 23.68 ± 3.04 | 23.80 ± 2.79 | −0.148 | 0.883 |
| PL (Left/right) | 4/25 | 2/24 | 0.525 | 0.469 |
| SL (Left/right) | 17/12 | 19/7 | 1.267 | 0.260 |
| SKES (Nm) | 73.16 ± 29.52 | 78.05 ± 23.95 | −0.669 | 0.506 |
| WKES (Nm) | 67.30 ± 28.68 | 55.16 ±19.23 | 1.823 | 0.074 |
| Asymmetry (%) | 8.97 ± 4.72 | 29.78 ± 8.64 | −11.246 | <0.001 |

Notes:
  BMI, body mass index; PL, preferred limb; SL, strong limb; SKES, strong knee extension strength; WKES, weak knee extension strength.

**Table 2 Indicators involved in gait test.**

| Parameters | SG | AG | $t$ | $P$ |
|---|---|---|---|---|
| Gait velocity | 92.52 ± 16.31 | 77.00 ± 15.78 | 3.577 | 0.001** |
| Cadence | 100.44 ± 18.75 | 89.88 ± 13.53 | 2.371 | 0.021* |

Notes:
  * $P < 0.05$,
  ** $P < 0.01$.

the two participant groups. As shown in Table 3, there was a significant intergroup main effect of step length ($F_{1,53} = 10.555$, $P < 0.01$, $\eta_p^2 = 0.166$), with *post hoc* analysis revealing that the step lengths of both strong and weak legs were significantly shorter in AG than in SG ($P < 0.01$ and $P < 0.01$, respectively). A significant intergroup main effect was observed in stride length ($F_{1,53} = 6.849$, $P < 0.05$, $\eta_p^2 = 0.114$); *post hoc* analysis indicated that both the strong and weak legs stride lengths were significantly shorter in AG than in SG ($P < 0.05$ and $P < 0.05$, respectively). In terms of step time, there was a significant intergroup main effect ($F_{1,53} = 19.278$, $P < 0.001$, $\eta_p^2 = 0.267$), with *post hoc* analysis revealing that both strong and weak legs step times were significantly longer in AG than in SG ($P < 0.05$ and $P < 0.001$, respectively). We found a significant intergroup main effect of stride time ($F_{1,53} = 10.303$, $P < 0.01$, $\eta_p^2 = 0.163$); *post hoc* analysis indicated that both strong and weak legs stride times were significantly longer in AG than in SG ($P < 0.05$ and $P < 0.01$, respectively). There was a significant intergroup main effect of stance time ($F_{1,53} = 22.420$, $P < 0.001$, $\eta_p^2 = 0.297$); *post hoc* analysis showed that both strong and weak legs stance times were significantly longer in AG than in SG ($P < 0.05$ and $P < 0.05$, respectively); however, strong leg stance time was significantly shorter than weak leg stance time among AG subjects ($P < 0.05$). We observed significant intergroup main effects of swing time ($F_{1,53} = 5.130$, $P < 0.05$, $\eta_p^2 = 0.088$); *post hoc* analysis revealed that the weak leg swing time in AG was significantly longer than that of SG ($P < 0.05$). SST demonstrated a significant intergroup main effect ($F_{1,53} = 5.012$, $P < 0.05$, $\eta_p^2 = 0.086$); *post hoc* analysis

**Table 3 The parameters of two-way repeated-measures ANOVA during gait test.**

| Gait parameter | SG Strong leg | SG Weak leg | AG Strong leg | AG Weak leg | Intragroup effect F | P | $\eta_p^2$ | Interaction effect F | P | $\eta_p^2$ | Intergroup effect F | P | $\eta_p^2$ |
|---|---|---|---|---|---|---|---|---|---|---|---|---|---|
| **Spatial parameters** | | | | | | | | | | | | | |
| Step length (cm) | 61.41 ± 8.05 | 60.13 ± 8.52 | 54.73 ± 6.46 | 54.33 ± 7.17 | 1.237 | 0.271 | 0.023 | 0.335 | 0.565 | 0.006 | 10.555 | 0.002** | 0.166 |
| Stride length (cm) | 118.44 ± 16.93 | 118.07 ± 15.15 | 108.00 ± 13.85 | 107.84 ± 13.06 | 0.121 | 0.729 | 0.002 | 0.018 | 0.893 | 0.000 | 6.849 | 0.012* | 0.114 |
| **Temporal parameters** | | | | | | | | | | | | | |
| Step time (s) | 0.56 ± 0.08 | 0.56 ± 0.06 | 0.64 ± 0.11 | 0.64 ± 0.09 | 0.001 | 0.977 | 0.000 | 0.215 | 0.645 | 0.004 | 19.278 | 0.000*** | 0.267 |
| Stride time (s) | 1.16 ± 0.21 | 1.17 ± 0.14 | 1.30 ± 0.16 | 1.29 ± 0.17 | 0.016 | 0.898 | 0.000 | 0.034 | 0.855 | 0.001 | 10.303 | 0.002** | 0.163 |
| Stance time (s) | 0.67 ± 0.08 | 0.67 ± 0.07 | 0.79 ± 0.13 | 0.81 ± 0.13 | 2.169 | 0.147 | 0.039 | 2.726 | 0.105 | 0.049 | 22.420 | 0.000*** | 0.297 |
| Swing time (s) | 0.43 ± 0.09 | 0.41 ± 0.05 | 0.45 ± 0.05 | 0.44 ± 0.06 | 0.972 | 0.329 | 0.018 | 0.000 | 0.996 | 0.000 | 5.130 | 0.028* | 0.088 |
| Single support time (s) | 0.45 ± 0.06 | 0.45 ± 0.06 | 0.47 ± 0.05 | 0.49 ± 0.05 | 1.928 | 0.171 | 0.035 | 1.901 | 0.174 | 0.035 | 5.012 | 0.029* | 0.086 |
| Double support time (s) | 0.22 ± 0.03 | 0.22 ± 0.03 | 0.31 ± 0.08 | 0.31 ± 0.08 | 0.223 | 0.639 | 0.004 | 0.054 | 0.817 | 0.001 | 31.794 | 0.000*** | 0.375 |
| **Temporophasic parameters** | | | | | | | | | | | | | |
| Stance time (%GC) | 61.47 ± 2.70 | 61.85 ± 1.29 | 63.39 ± 2.45 | 64.31 ± 3.92 | 1.797 | 0.186 | 0.033 | 0.314 | 0.578 | 0.006 | 15.584 | 0.000*** | 0.227 |
| Swing time (%GC) | 38.53 ± 2.70 | 38.16 ± 1.29 | 36.61 ± 2.45 | 35.69 ± 3.92 | 1.797 | 0.186 | 0.033 | 0.314 | 0.578 | 0.006 | 15.584 | 0.000*** | 0.227 |
| Single support time (%GC) | 38.73 ± 3.04 | 38.44 ± 2.68 | 36.39 ± 3.79 | 38.05 ± 4.18 | 0.720 | 0.400 | 0.013 | 1.484 | 0.229 | 0.027 | 8.530 | 0.005** | 0.139 |
| Double support time (%GC) | 19.31 ± 3.29 | 18.99 ± 2.73 | 23.67 ± 4.31 | 23.45 ± 6.19 | 0.195 | 0.661 | 0.004 | 0.008 | 0.928 | 0.000 | 20.412 | 0.000*** | 0.278 |

**Notes:**
* $P < 0.05$,
** $P < 0.01$
*** $P < 0.001$.

indicated that weak leg SST was significantly greater in AG than in SG ($P < 0.05$). A significant intergroup main effect was noted for DST ($F_{1,53} = 31.794$, $P < 0.001$, $\eta_p^2 = 0.375$), and *post hoc* analysis revealed that both strong and weak legs DST was significantly greater in AG than in SG ($P < 0.05$ and $P < 0.05$, respectively). STT%, exerted significant intragroup ($F_{1,53} = 8.799$, $P < 0.001$, $\eta_p^2 = 0.142$). *Post hoc* analysis showed that weak side STT% were significantly greater than in AG than in SG ($P < 0.05$). We observed significant intergroup main effects of SWT% ($F_{1,53} = 15.584$, $P < 0.001$, $\eta_p^2 = 0.227$). *Post hoc* analysis indicated that SWT% for both strong and weak sides were significantly smaller in AG than in SG ($P < 0.01$ and $P < 0.01$, respectively). A significant intergroup main effect was observed for SST% ($F_{1,53} = 8.530$, $P < 0.01$, $\eta_p^2 = 0.139$); *post hoc* analysis showed a significantly smaller strong leg SST% in AG compared to SG ($P < 0.05$). DST% demonstrated was a significant intergroup main effect ($F_{1,53} = 20.412$, $P < 0.001$,

$\eta_p^2 = 0.278$); *post hoc* analysis indicated that DST% for both strong and weak sides was significantly greater in AG than in SG ($P < 0.001$ and $P < 0.01$, respectively).

## Static postural control

ANOVA was performed to assess the differences in static postural control between two groups, with SG and AG as between-group factors and comfortable standing test and narrow stance test as within-group factors. The analysis aimed to investigate the variations across wide and narrow stance test tasks. Data are reported in Table 4, there was a significant intragroup main effect on EO-SVI ($F_{1,53} = 14.419$, $P < 0.001$, $\eta_p^2 = 0.214$), and NSEO-SVI was significantly greater than CSEO-SVI ($P < 0.01$). There was a significant intergroup main effect of EO-SVI ($F_{1,53} = 14.223$, $P < 0.001$, $\eta_p^2 = 0.212$); simple effect analysis indicated that under both standing distance conditions, EO-SVI significantly higher in AG than in SG ($P < 0.05$ and $P < 0.01$, respectively); however, NSEO-SVI was significantly greater than CSEO-SVI ($P < 0.001$). EC-SVI demonstrated inter-and intragroup interaction effects ($F_{1,53} = 8.137$, $P < 0.01$, $\eta_p^2 = 0.133$); *post hoc* analysis revealed that under both standing distance conditions, EC-SVI was significantly greater in AG than in SG ($P < 0.01$ and $P < 0.001$, respectively). However, within SG, there was no significant difference ($P > 0.05$). Within AG, NSEC-SVI was significantly greater than CSEC-SVI ($P < 0.001$). Neither the main nor interaction effects of EO-AOE were significant ($P > 0.05$). *Post hoc* analysis revealed that under the NS condition, EO-AOE was significantly lower in SG compared to AG ($P < 0.05$). There were significant inter-and intragroup interactions effects on EC-AOE ($F_{1,53} = 11.342$, $P < 0.01$, $\eta_p^2=0.176$). Simple effect analysis indicated that under both testing conditions, EC-AOE was significantly larger in AG than in SG ($P < 0.01$ and $P < 0.001$, respectively). However, there was no significant difference within SG ($P > 0.05$). Within AG, NSEC-AOE was significantly larger than CSEC-AOE ($P < 0.001$).

## Correlation between static postural control and gait performance

Correlations between static postural control and gait performance were examined through Pearson correlation analysis. As shown in Fig.1. In the CSEO condition, the SVI exhibited negative correlations with strong side stride length ($r = -0.298$, $P = 0.027$), weak side stride length ($r = -0.343$, $P = 0.010$). In the CSEC condition, SVI was negatively correlated with strong side step length ($r = -0.301$, $P = 0.026$); and weak side swing time ($r = -0.295$, $P = 0.029$); however, SVI showed positive correlations with strong and weak sides stance time ($r = 0.302$, $P = 0.025$; $r = 0.328$, $P = 0.014$); strong and weak side DST ($r = 0.340$, $P = 0.011$; $r = 0.345$, $P = 0.010$); weak side STT% ($r = 0.295$, $P = 0.029$); and weak side DST% ($r = 0.271$, $P = 0.046$). In the NSEO condition, SVI positively correlated with strong side step time ($r = 0.440$, $P = 0.001$); strong and weak sides stance time ($r = 0.461$, $P = 0.000$; $r = 0.449$, $P = 0.001$); strong and weak sides DST ($r = 0.423$, $P = 0.001$; $r = 0.478$, $P = 0.000$); strong and weak sides STT% ($r = 0.459$, $P = 0.000$; $r = 0.428$, $P = 0.001$); and strong and weak sides DST% ($r = 0.387$, $P = 0.004$; $r = 0.331$, $P = 0.014$). However, within NSEO condition, SVI negatively correlated with strong and weak sides SWT% ($r = -0.459$, $P = 0.000$; $r = -0.428$, $P = 0.001$). In the NSEC condition, SVI correlated negatively with

**Table 4 The parameters of two-way repeated-measures ANOVA during static postural control test.**

| | SG | | AG | | Intragroup effect | | | Interaction effect | | | Intergroup effect | | |
|---|---|---|---|---|---|---|---|---|---|---|---|---|---|
| Parameters | CS | NS | CS | NS | F | P | $\eta_p^2$ | F | P | $\eta_p^2$ | F | P | $\eta_p^2$ |
| EO-SVI | 7.77 ± 1.83 | 8.25 ± 1.82 | 9.31 ± 2.79 | 10.85 ± 2.56 | 14.419 | 0.000*** | 0.214 | 3.963 | 0.052 | 0.070 | 14.223 | 0.000*** | 0.212 |
| EC-SVI | 7.81 ± 1.61 | 8.29 ± 1.28 | 9.80 ± 2.90 | 11.74 ± 2.23 | 22.101 | 0.000*** | 0.294 | 8.137 | 0.006** | 0.133 | 30.207 | 0.000*** | 0.363 |
| EO-AOE | 115.89 ± 110.41 | 110.41 ± 82.65 | 142.14 ± 107.43 | 177.55 ± 113.92 | 1.274 | 0.264 | 0.023 | 2.378 | 0.129 | 0.043 | 3.560 | 0.065 | 0.063 |
| EC-AOE | 89.02 ± 68.97 | 108.93 ± 44.89 | 162.95 ± 99.50 | 267.81 ± 156.13 | 24.461 | 0.000*** | 0.316 | 11.342 | 0.001** | 0.176 | 24.152 | 0.000*** | 0.313 |

**Notes:**

CS, comfortable stance; NS, narrow stance; EO, eyes open; EC, eyes close; SVI, sway velocity index; AOE, area of ellipse.

\* $P < 0.05$

\*\* $P < 0.01$

\*\*\* $P < 0.001$.

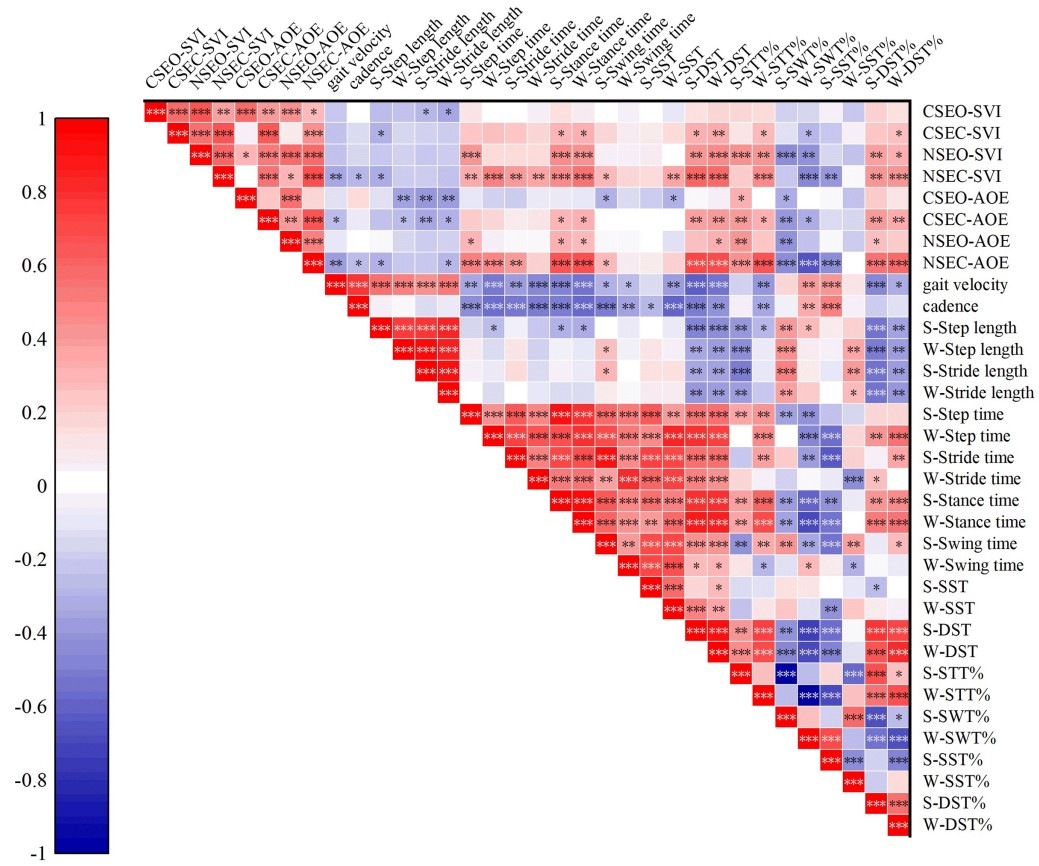

**Figure 1 Correlations among static postural control and gait test.** CSEO, comfortable stance with eyes open; CSEC, comfortable stance with eyes close; NSEO, narrow stance with eyes open; NSEC, narrow stance with eyes close; S, strong limb; W, weak limb. \*$P < 0.05$; \*\*$P < 0.01$; \*\*\*$P < 0.001$.

gait velocity (r = −0.350, P = 0.009); cadence (r = −0.281, P = 0.038); strong side step length (r = −0.322, P = 0.016); weak side SWT% (r = −0.469, P = 0.000); and strong side SST% (r = −0.415, P = 0.002); however, SVI exhibited positive correlations with strong and weak sides step time (r = 0.345, P = 0.01; r = 0.519, P = 0.000); strong and weak sides stride time (r = 0.417, P = 0.002; r = 0.373, P = 0.005); strong and weak sides stance time (r = 0.477,

$P = 0.000$; r = 0.546, $P = 0.000$); strong side swing time (r = 0.298, $P = 0.027$); weak side SST (r = 0.376, $P = 0.005$); strong and weak sides DST (r = 0.609, $P = 0.000$; r = 0.587, $P = 0.000$); weak side STT% (r = 0.469, $P = 0.000$); and strong and weak sides DST% (r = 0.430, $P = 0.001$; r = 0.437, $P = 0.001$). In the CSEO condition, AOE showed negative correlations with weak side step length (r = −0.377, $P = 0.005$); strong and weak sides stride length (r = −0.419, $P = 0.001$; r = −0.388, $P = 0.033$); strong side swing time (r = −0.313, $P = 0.020$); weak side SST (r = −0.301, $P = 0.025$); and strong side SWT% (r = −0.321, $P = 0.017$). However, in the CSEO condition, AOE showed positive correlations with strong side STT% (r = 0.321, $P = 0.017$). In the CSEC condition, AOE exhibited negative correlations with gait velocity (r = −0.273, $P = 0.044$); weak side step length (r = −0.272, $P = 0.045$); strong and weak sides stride length (r = −0.349, $P = 0.009$; r = −303, $P = 0.025$); and strong and weak sides SWT% (r = −0.405, $P = 0.002$; r = −0.322, $P = 0.013$); however, in the CSEC condition, AOE exhibited positive correlations with strong and weak sides stance time (r = 0.303, $P = 0.025$; r = 0.321, $P = 0.017$); strong and weak sides DST (r = 0.390, $P = 0.003$; r = 0.415, $P = 0.002$); strong and weak sides STT% (r = 0.405, $P = 0.002$; r = 0.332, $P = 0.013$); and strong and weak sides DST% (r = 0.415, $P = 0.002$; r = 0.359, $P = 0.007$). In the NSEO condition, AOE positively correlated with strong side step time (r = 0.301, $P = 0.026$); strong and weak sides stance time (r = 0.328, $P = 0.014$; r = 0.288, $P = 0.033$); weak side DST (r = 0.320, $P = 0.017$); strong side SST% (r = 0.423, $P = 0.001$); and strong side DST% (r = 0.266, $P = 0.050$); however, in the NSEO condition, AOE negatively correlated with strong side SWT% (r = −0.423, $P = 0.001$). In the NSEC condition, AOE negatively correlated with gait velocity (r = −0.387, $P = 0.004$); cadence (r = −0.279, $P = 0.039$); strong side step length (r = −0.315, $P = 0.019$); weak side stride length (r = −0.285, $P = 0.035$); strong and weak SWT% (r = −0.459, $P = 0.000$; r = −0.669, $P = 0.000$); and strong side SST% (r = −0.453, $P = 0.001$). However, in the NSEC condition, AOE exhibited positive correlations with strong and weak sides step time (r = 0.508, $P = 0.000$; r = 0.498, $P = 0.000$); strong stride time (r = 0.411, $P = 0.002$); strong and weak sides stance time (r = 0.669, $P = 0.000$; r = 0.692, $P = 0.000$); strong swing time (r = 0.283, $P = 0.037$); strong and weak DST (r = 0.722, $P = 0.000$; r = 0.709, $P = 0.000$); strong and weak sides STT% (r = 0.459, $P = 0.000$; r = 0.669, $P = 0.000$); and strong and weak sides DST% (r = 0.543, $P = 0.000$; r = 0.592, $P = 0.000$).

## DISCUSSION

This study assessed how lower limb muscle strength asymmetry impacts gait and postural control in older adults. Using natural walking and upright standing tests, we found that the AG exhibited more pronounced abnormalities in gait parameters, with the weaker side showing greater deviations. AG also demonstrated poorer static postural control, except for CSEO-AOE. What counts was poorer static postural control correlated with decreased gait efficiency, indicating that static postural control is vital in the context of muscle strength asymmetry's effects on gait performance.

The strong correlation between walking speed and muscle strength has been widely recognised in the scientific literature (*Hayashida et al., 2014*; *Kanayama et al., 2022*). Our study found that the gait velocity and cadence were considerably lower in AG

compared to SG. This supports the findings of *Portegijs et al. (2005)*. This may be due to the fact that in the AG, the weaker limb's muscle strength is insufficient to meet the demands of the overall motion. Attaining a specified degree of muscular strength is crucial for executing motions (*Yoshioka et al., 2007*); if muscular strength does not meet the requirements of a given action, the execution of that action will be hindered (*Bean et al., 2002*; *Santos et al., 2022*). The gait cycle during natural walking speed can be divided into two phases: the stance phase, which accounts for 60% of the cycle, and the swing phase, which accounts for the remaining 40% (*Murray, Drought & Kory, 1964*). Furthermore, our investigation revealed a larger STT% and smaller SWT% on the weak side in the AG, which can be attributed to impaired movement patterns and posture resulting from either excessive or insufficient muscular activation (*Trulsson et al., 2015*). Asymmetrical muscle strength disrupted the timing of muscle activation during walking, hence impacting proprioceptive feedback from the joints (*Shakoor et al., 2014*). Although compensatory mechanisms and the restoration of muscle balance can partially alleviate aberrant gait (*Reid et al., 2014*), surpassing the body's threshold for muscle imbalance can result in visible gait disorders (*Gardinier et al., 2014*). Additionally, our analysis indicates that the weak limb exhibits an even longer stance time than the strong limb, a detail not highlighted in prior research. These phenomena can be attributed to the disparity in the forces that provide support and propulsion induced by the asymmetrical lower limb muscle strength (*Laroche, Cook & Mackala, 2012*). This extended stance time is further influenced by the uneven distribution of joint loading and the subsequent compensatory mechanisms (*Bond et al., 2017*; *Shi et al., 2019*). The strength asymmetry means the weak leg cannot support the same load as the strong leg, causing the body's center of gravity to shift towards the weak side. To maintain balance, the weak leg increases its contact time with the ground, thereby generating a greater ground reaction force to offset its reduced strength, as evidenced by the observed increase in weak leg stance time in this study. This gait pattern is likewise applicable to the increase of SST% and DST%. In summary, lower limb muscle strength asymmetry has a substantial impact on the gait performance of older individuals, specifically in AG individuals, where more pronounced abnormalities of gait parameters are observed in the weak side.

This study examined the influence of lower limb muscle strength asymmetry on maintaining stable body posture in both CS and NS conditions, with eyes either open or closed. To begin, three factors can be identified as the origins of the intergroup differences in NSEO-SVI, NSEC-SVI, NSEO-AOE, and NSEC-AOE. To begin with, static erect posture instability in the medial-lateral and anterior-posterior directions during standing arises from unilateral muscle weakness, which predisposes to fatigue (*Manty et al., 2012*). *Paillard & Borel (2013)* have verified that muscle exhaustion, whether unilateral or bilateral, increases the pace of sideways swaying and alters the trajectory of the centre of pressure in the anterior-posterior and medial-lateral directions, resulting in asymmetrical posture. Furthermore, asymmetrical lower limb muscle strength can impair the intrinsic biomechanical properties of the lower limb and alter proprioceptive information crucial for postural control (*Scrivens, DeWeerth & Ting, 2008*; *Wang et al., 2022*). Lastly, combined with insufficient unilateral muscle strength, the dual difficulties of maintaining a

narrow posture and losing visual acuity further burden elderly individuals, ultimately compromising postural control. In contrast to the study hypothesis, no significant intergroup difference was found in CSEO-AOE. This could be attributed to the directional elements incorporated into the study metrics. SVI is linked to the medial-lateral movement of the centre of mass, whereas AOE is not only connected to medial-lateral movement but also to anterior-posterior movement of the centre of mass. Ultimately, only the AG group exhibited increases in both EC-SVI and EC-AOE in the NS condition when compared to the CS posture. *Palmer, Farrow & Palmer (2020)* discovered a direct correlation between muscle mass (echo intensity) and sway index when the eyes are closed in older individuals. This phenomenon may be attributed to the isolation of visual conditions that sustain balance, as well as the more demanding standing posture, which imposes greater demands on postural control (*Macedo et al., 2015*), particularly on the initially weak side.

To summarise, an imbalance in the muscle strength of the lower limbs compromises the ability of older adults to maintain balance in a stationary position when standing with their feet close together and their eyes closed.

The relationship between static postural control and gait performance was even more thoroughly examined in this study. A significant correlation between gait performance and the static posture control test metrics was observed across all conditions. Furthermore, under condition NSEC, significant association was observed between both SVI and AOE with the majority of gait parameters. Specifically, a decrease in static postural control was associated with a higher degree of aberrant gait performance. Inconsistent muscle strength causes unilateral fatigue and disrupts proprioception (*Manty et al., 2012*; *Zeng et al., 2022*), subsequently disrupting the balance and control of lower limbs movements (*Konstantopoulos et al., 2021*). Such instability on the medial-lateral and anterior-posterior planes while standing can exacerbate the fluctuations in the centre of mass during walking, culminating in an irregular gait pattern. Additionally, the associations between static posture control and gait parameters such as step time, stance time, DST, and DST% did not exhibit lateralization. This finding aligns with the previous research (*Callisaya et al., 2010*), which noted that individuals with unstable static posture compensate for inefficient gait control by increasing the duration of double-foot-ground contact (*Arpan et al., 2022*). Considering that gait is an uninterrupted sequence of movements, any disruptions in postural control can adversely affect the biomechanics of the lower limbs (*Scrivens, DeWeerth & Ting, 2008*; *Shanbhag et al., 2023*). Different from previous studies, our study uncovers a significant association between weakened static posture control and a reduction in SST% on the strong leg. This finding also implies that individuals with impaired posture control are more likely to display an elevated SST% on their weak limb. Our novel findings are supported by several factors. On one hand, the fatigue susceptibility of the weak leg can impact overall balance and gait performance (*Manty et al., 2012*; *Arora et al., 2015*). On the other hand, the weak leg is susceptible to causing an uneven distribution of load on the knee, leading to a shift in the body's center of gravity towards the weak side (*Shi et al., 2019*), which in turn affects the stability of the standing posture and the natural fluidity of gait. Finally, the quadriceps strength of the weak leg influences proprioceptive feedback adjustments, leading to altered joint position perception and improper postural

adjustments (*Zeng et al., 2022*). This is compounded by the fact that, during walking, the weak limb may compensate for its reduced strength by increasing muscle activation time. The continuous ground contact by the weak leg generates reactive forces, which meet the demands of walking (*Bond et al., 2017*). These adaptations ultimately result in an abnormal gait pattern.

To our knowledge, this study is the first investigation of the effect of static postural control on the connection between lower limb muscle strength asymmetry and gait performance in older individuals. There are two findings that hold potential implications for future clinical practices. Firstly, when interlimb strength asymmetry exceeds 15%, the gait performance of the weaker limb tends to be more abnormal. Interventions can target this characteristic by employing unilateral muscle strength training to improve the strength of the weaker limb (*Appleby, Cormack & Newton, 2019*). Secondly, the deterioration of static postural control due to asymmetry can further impair the gait performance of older adults. This necessitates a focus on enhancing joint and muscle control capabilities during exercise interventions, such as those involving neuromuscular training. This type of training can optimize cortical input signals and information integration, thereby strengthening involuntary motor responses that are crucial for joint control (*Risberg et al., 2001*). Nevertheless, limitations of this study must be acknowledged. To begin with, its cross-sectional study design does not permit causal inferences. To further validate the results obtained in this study, future investigations should incorporate randomised controlled trials. Furthermore, this study employed the widely accepted 15% threshold as a preset criterion for asymmetry. Several researchers contend that the use of pre-established thresholds should be avoided while examining asymmetry-related disorders (*Parkinson et al., 2021*). Subsequent investigations should include comparative analyses of various thresholds of asymmetry to evaluate disparities in gait performance, with the objective of identifying the most effective approach for categorising asymmetry. Finally, this study evaluated lower limb muscle strength asymmetry by measuring the discrepancy in bilateral knee extension strength. While this approach is frequently employed in contemporary studies to measure the imbalance in lower limb muscle strength (*Straight, Brady & Evans, 2016*), it is important to note that human walking engages several joints in the lower limb. Subsequent research should incorporate strength imbalances between the hip, knee, and ankle joints to assess their influence on walking patterns.

## CONCLUSIONS

1) Elderly individuals with asymmetric lower limb muscle strength exhibit altered gait patterns, including slower velocity, lower cadence, reduced step and stride lengths, prolonged step and stance times, a decrease in weak leg swing time, an extension of the weak leg single support time, and an increase in double support time. Additionally, there is a notable increase in the proportion of the gait cycle dedicated to stance and swing phases, along with an elevated percentage of the strong leg single support and double support times. It should be noted that the stance time of the weak leg is longer when compared to the strong leg.

2) Lower limb muscle strength asymmetry impedes static postural control in older adults, manifested by increased medio-lateral swing velocity and larger anterior-posterior and medio-lateral pressure centre movement trajectories. In particular, stability in the medio-lateral direction was impaired.

3) The mechanism underlying the abnormal gait resulting from lower limb muscle strength asymmetry in older individuals may be attributed to compromised static postural control.

### Funding
This research was supported by grants from The Program for Overseas High-level talents at Shanghai Institutions of Higher Learning (No. TP2020063). The funders had no role in study design, data collection and analysis, decision to publish, or preparation of the manuscript.

### Grant Disclosures
The following grant information was disclosed by the authors:
Shanghai Institutions of Higher Learning: TP2020063.

### Competing Interests
The authors declare that they have no competing interests.

### Author Contributions
- Beili Si conceived and designed the experiments, performed the experiments, analyzed the data, prepared figures and/or tables, authored or reviewed drafts of the article, and approved the final draft.
- Hao Zhu conceived and designed the experiments, analyzed the data, authored or reviewed drafts of the article, and approved the final draft.
- Xinmei Wei performed the experiments, prepared figures and/or tables, recruited the subjects, and approved the final draft.
- Shun Li performed the experiments, prepared figures and/or tables, and approved the final draft.
- Xueping Wu conceived and designed the experiments, authored or reviewed drafts of the article, and approved the final draft.

### Human Ethics
The following information was supplied relating to ethical approvals (*i.e.*, approving body and any reference numbers):

Ethics Committee of Shanghai University of Sport approved this research (No.102772022RT123)

### Data Availability
The raw measurements are available in the Supplemental File.

## Supplemental Information

Supplemental information for this article can be found online at http://dx.doi.org/10.7717/peerj.17626#supplemental-information.

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
