# Peer review of "The mechanism of static postural control in the impact of lower limb muscle strength asymmetry on gait performance in the elderly"

_PeerJ, doi:10.7717/peerj.17626_

## Round 0.1 · original submission · Major Revisions

Dear Co-Authors:

Thank you for submitting your manuscript to PeerJ Journals. Your manuscript has been reviewed, and we need to make some improvements to consider it for publication. Please attend to the reviewers' comments.

Dr. Manuel Jimenez

·

Basic reporting

This article reports data of a cross-sectional study investigating correlations between lower limb muscle strength asymmetry, postural control and gait performance. It is written in a clear and unambigous, professional English. The literature references are provided, however I think that the following papers exploring similar research questions or demonstrating validity of wearable sensors in detecting gait performance would provide strong additional context information: line 43 REF "associations of Gait Disorders and recurrent falls in older people: a prospective population-based study, Marini et al., Gerontology 2021; line 47 REF "Instrumented gait analysis defines the walking signature of CACNA1A disorders, Indelicato et al., 2022, J of Neurology". Since an association between orthostatic hypotension and gait performance has been described, please cite "the footprint of orthostatic hypotension in parkinsonian syndromes, Park and Rel Dis 2020 Raccagni et al.
Figures and Tables are well designed, raw data are shared. The article is well-contained with relevant results to hypotheses.

Experimental design

The explored research area is novel, the research question is well defined and relevant. The methods are described in detail, ethical standard reflect a rigorous investigation.

Validity of the findings

No comment.

·

Basic reporting

In general the manuscript is well written and clear. There are a few minor instances where sentences or structure should be improved for clarity. For example:
-The second sentence of the 1st paragraph of the introduction (lines 44-47) has semi-colons instead of commas
-The sentence beginning 'Postural control is related...' (Line 70) is an awkward sentence and should be re-written
-The content from lines 90-112 appear to be methods-based and would be better suited in a more concise form in the methods section.
-Please check the reporting of the p-values throughout the results section (i.e. state exact p value if between 0.001 and 0.05; else use <0.001, >0.05)
I also think the background/rationale for using the knee extensor torques as tested is not made evident in the introduction section -- I strongly suggest adding this.

Experimental design

Generally I think this is a well-designed study with a very interesting and useful research question. The methods as written are clear, however I do have a few questions.
1. Why was only the knee extension force (torque) used to measure leg muscle imbalances? Given the importance of the ankle strategy (and thus the plantar/dorsiflexors') primary involvement in controlling balance in quiet stance, the inclusion of this muscle group is not clear. Moreover, given the first metric of sway includes only mediolateral displacement, I question the muscle group choice.
2. Why were 5 maximal tests performed? How were these data utilised to compute asymmetry? Was it the average of all 5 trials? Or just the peak value from one?
3. Why was AP sway not included as a metric on its own? The SVI looks at mediolateral, and the AOE includes both, but AP sway is not reported alone.

Validity of the findings

I believe the data in general have been analysed appropriately. However, my major concern is that the way in which the gait parameters have been analysed, and thus reported, focus on left vs right leg, rather than strong vs weak leg, when discussing imbalances. As not all participants demonstrated the same stronger leg, I think this is disingenuous to report the LvsR imbalances and form the discussion around this comparison. As such, I would like to see the statistical analyses re-done to focus on strong vs weak leg. This may also necessitate a re-write of the discussion/conclusion section.

---

## Round 0.2 · Minor Revisions

Dear Authors:

Please address the comments of Reviewer 3

Thank you

Dr. Manuel Jiménez

·

Basic reporting

no comment

Experimental design

The statistical analysis has been reorganised to include strong vs weak leg comparisons and is much clearer now.

Validity of the findings

no comment

Additional comments

Thank you for addressing my (and the other reviewer's) comments and for the very succinct responses. The manuscript is much improved. Best of luck with the PhD completion - I look forward to seeing further outputs from this work.

·

Basic reporting

all contents regards to basic reporting were considered in this version of the manuscript.

Experimental design

good

Validity of the findings

excellent.

Additional comments

in the method section the paper need sample size calculation. and additional inclusion and exclusion criteria.

discussion is very long and boring. please reduce the words in this section. in addition add clinical view of your findings in this section.

---

## Round 0.3 · accepted · Accept

Dear authors

Thank you for submitting your manuscript to PeerJ and for your response to reviewer comments. After reading the final manuscript, I consider that the improvements have been substantial and that it meets the necessary quality to be published in our journal.

Congratulations on your work and for choosing PeerJ.

A cordial greeting.

·

Basic reporting

All my comments well done.

Experimental design

All my comments well done.

Validity of the findings

All my comments well done.

Additional comments

No